# In-Depth Understanding of *Ecklonia stolonifera* Okamura: A Review of Its Bioactivities and Bioactive Compounds

**DOI:** 10.3390/md20100607

**Published:** 2022-09-27

**Authors:** Xiao Men, Xionggao Han, Se-Jeong Lee, Geon Oh, Heegu Jin, Hyun-Ji Oh, Eunjin Kim, Jongwook Kim, Boo-Yong Lee, Sun-Il Choi, Ok-Hwan Lee

**Affiliations:** 1Department of Food Biotechnology and Environmental Science, Kangwon National University, Chuncheon 24341, Korea; 2Department of Food Science and Biotechnology, College of Life Science, CHA University, Seongnam 13488, Korea; 3Naturalway Co., Ltd., Pocheon 11160, Korea

**Keywords:** *Ecklonia stolonifera* Okamura, phytochemistry, bioactive compound, bioactivity, health benefit

## Abstract

*Ecklonia stolonifera* Okamura (ES) is mainly distributed in the coastal areas of the middle Pacific, around Korea and Japan, and has a long-standing edible value. It is rich in various compounds, such as polysaccharides, fatty acids, alginic acid, fucoxanthin, and phlorotannins, among which the polyphenol compound phlorotannins are the main active ingredients. Studies have shown that the extracts and active components of ES exhibit anti-cancer, antioxidant, anti-obesity, anti-diabetic, antibacterial, cardioprotective, immunomodulatory, and other pharmacological properties in vivo and in vitro. Although ES contains a variety of bioactive compounds, it is not widely known and has not been extensively studied. Based on its potential health benefits, it is expected to play an important role in improving the nutritional value of food both economically and medically. Therefore, ES needs to be better understood and developed so that it can be utilized in the development and application of marine medicines, functional foods, bioactive substances, and in many other fields. This review provides a comprehensive overview of the bioactivities and bioactive compounds of ES to promote in-depth research and a reference for the comprehensive utilization of ES in the future.

## 1. Introduction

*Ecklonia stolonifera* Okamura (ES) is a perennial benthic brown alga widely distributed in the middle Pacific coastal areas of Korea and Japan and belongs to the family Laminariaceae, order Laminariales [1,2]. It has strong reproductive ability and can adapt well to environmental changes. ES usually grows in the sea at a depth of 2–10 m [3]. The erect thalli length of ES is related to the underwater growth depth, and the length of ES grown, at a depth of 2 m, is significantly longer than that of ES at 1.0–1.5 m and 2.5–3.0 m [4]. The leaves can grow from 45.2 to 96.5 cm in length and 13.7 to 28.6 cm in width. The length and width of leaves are affected by different months; the leaves are longer from March to July, and their width is narrower in November. ES is an economical seaweed that is widely used in food and industry [5]. In the abalone farming industry, ES is considered an excellent candidate for abalone feed. Compared with other brown algae, ES can continue to grow when the water temperature increases in summer, providing a rich food source for the growth of abalone. Moreover, the shell color of farmed abalone can be more natural dark brown, which is more popular in the market [6]. In recent years, many studies have shown that ES has great development potential in functional foods and pharmaceuticals, and its extract contains a high content of phenolic compounds, showing high antioxidant capacity, including strong DPPH free radical-scavenging activity, ferrous reducing power, and superoxide radical-scavenging activity [7]. ES has a variety of biological activities and has attracted much attention because of its high concentration of polysaccharides, fucosterol, fatty acids, phlorotannins of eckol, diekcol, phlorofucofuroeckol A, eckstolonol, and phenolic compounds with antioxidant capacities [7,8,9]. Through gel filtration column and high-performance liquid chromatography (HPLC) analysis, Kuda et al. determined that the water extract of ES contained low molecular weight polysaccharides and that these polysaccharides were laminarin [7]. The results of sugar composition analysis indicated that the types of polysaccharides contained in ES cell walls were cellulose, fucoidan, and laminaran [10].

Marine brown algae contain a variety of phloroglucinol-based polyphenols, among which phlorotannins are a group of phenolic compounds restricted to phloroglucinol polymers [11,12]. It is the only phenolic group detected in brown marine algae [13,14]. These bioactive phenolic compounds, especially phlorotannins, have various beneficial biological activities and functions in marine brown algae. In previous studies, fourteen phlorotannins have been isolated and identified from ES: phloroglucinol, phlorofucofuroeckol A [15], phlorofucofuroeckol B [16], eckol [17], eckstolonol [15], triphlorethol-A [18], dieckol [15], fucofuroeckol-A [19], 7-resorcinol [20], 2-phloroeckol [16], 6,6′-bieckol [16], 974-A [21], 974-B [16], and dioxinodehydroeckol [22]. These bioactive substances enable ES to exhibit multiple biological functions, including antibacterial [23], tyrosinase-inhibiting [21,24], anti-photoaging [25], anti-hyperlipidemic [26], anti-inflammatory [22,23,27], anti-diabetic [27,28,29], and powerful antioxidant activities [1,21,22,29,30,31]. These health benefits and mechanisms of action were explored and demonstrated through in vivo and in vitro experiments. This review summarizes the latest knowledge on the composition, physicochemical and biological activities, and health benefits of ES.

## 2. Extraction Methods

The extraction and isolation of active ingredients in plants is a scientific, reasonable, and feasible extraction and isolation process designed according to the existence, solubility, and polarity of active ingredients in plants. The extraction and isolation of active ingredients are of great significance to improve the curative effect of drugs, expand medicinal plant resources, explore the therapeutic mechanism of plant active ingredients, and promote the research and development of new drugs and domestic medical undertakings [32]. The main extraction methods for extracting active ingredients from ES are summarized as follows (Table 1).

### 2.1. Methanol Extract

Jung et al. extracted 500 g of ES lyophilized powder with methanol to obtain 116.6 g of methanol extract, which was then fractionated and separated with dichloromethane (CH_2_Cl_2_), ethyl acetate (EtOAc), n-butanol (n-BuOH), water and silica gel chromatography column to obtain 1390 mg of fucosterol [2]. Moon et al. also performed reflux extraction of 500 g of lyophilized ES powder with methanol solvent and fractionated the methanol extract using the same fractionating solvent to analyze the most active EtOAc fraction. A total of six phlorotannins, 7-phloroeckol, dieckol, phlorofucofuroeckol A, eckol, dioxinodehydroeckol, and phloroglucinol were identified. The contents were 8 mg, 143 mg, 18 mg, 44 mg, 7 mg, and 28 mg, respectively [35]. Kim et al. isolated and identified three phlorotannins, dioxinodehydroeckol, dieckol, and phlorofucofuroeckol A from methanolic extracts of ES [22]. Kang et al. isolated five phlorotannins, dieckol, phlorofucofuroeckol A, eckol, eckstolonol, and phloroglucinol from the methanolic extract of lyophilized ES powder [24].

### 2.2. Ethanol Extract

Bang et al. carried out reflux extraction of 3 kg of air-dried ES powder with 70% ethanol solvent and determined by HPLC analysis that the contents of eckol, dieckol, and phlorofucofuroeckol-A in the ethanol extract were 12.59 ± 0.52 μg/g, 147.44 ± 2.27 μg/g, and 12.48 ± 1.06 μg/g [36]. In the study of Han et al., 70% ethanol was also used as the extraction solvent, and the content of dieckol in the ES ethanol extract was determined to be 27.42 ± 0.66 mg/g by HPLC analysis [38]. Two sterols, fucosterol (300 mg) and 24-hydroperoxy 24-vinylcholesterol (50 mg) were isolated and identified from the n-hexane level (27.9 g) of ES ethanol extract. Additionally, eight phlorotannins, 7-phloroeckol (20 mg), 2-phloroeckol (9 mg), triphlorethol-A (60 mg), dieckol (87 mg), phlorofucofuroeckol-A (57 mg), eckol (135 mg), eckstolonol (60 mg), and phloroglucinol (98 mg), were isolated from the EtOAc fraction (25.0 g) [18]. Wei et al. dried the collected ES under direct sunlight for 2 days, extracted the powder with 95% ethanol, and then dissolved the obtained powder extract with methanol. Eight kinds of phlorotannins were isolated: phlorofucofuroeckol A, dieckol, 974-B, 6,6′-bieckol, phlorofucofuroeckol B, eckol, dioxinodehydroeckol, and 2-phloroeckol [16]. Manandhar et al. extracted ES with 70% ethanol and obtained 63.8 g of EtOAc fraction from 200 g of ethanol extract, which was further separated by silica gel column. ES was determined to contain 24.5 mg of 974-A and three other phloroglucinol, eckol, and phlorofucofuroeckol-A phlorotannins [21].

### 2.3. Hexane Extract

Goo et al. extracted 50 g of ES lipid-removed dried powder by hexane solvent, the obtained extract was fractionated with n-BuOH, EtOAc, and methylene chloride, and the EtOAc fraction was analyzed by reversed phase-HPLC analysis. It was determined that the EtOAc fraction contained three major phlorotannins, phlorofucofuroeckol-A, dieckol, and eckol, and their contents were 17.7 ± 0.5 µg/mg, 30.1 ± 0.7 µg/mg, and 3.2 ± 0.2 µg/mg, respectively [37].

From the above summary, it can be seen that the types and contents of effective active ingredients in ES not only depend on the different extraction methods but also the different treatment methods of ES. The difference in active ingredient content may also depend on synergistic interactions between different active ingredients. Therefore, more kinds of effective active components could be expected to be isolated from ES and it can provide more information for elucidating the various components and contents contained in ES in future studies.

## 3. Bioactive Compounds

Numerous studies have shown that these bioactive compounds contained in ES are mainly phlorotannins, and sterols (Table 2). This also gives ES a variety of biological activities, such as anti-diabetic, antioxidant, anti-inflammatory, anti-obesity, etc. 

### 3.1. Fucosterol

Sterols are important natural active substances widely found in living organisms. They can be divided into three categories according to the source of raw materials: animal sterols, plant sterols, and fungal sterols [41]. Fucosterol is the main sterol metabolite in marine brown algae, accounting for 83–97% of sterol content [42]. Studies have shown that fucosterol in ES has anti-obesity and anti-diabetic effects [2,43,44]. In a previous study, the methanol extract of ES was further fractionated with CH_2_Cl_2_, EtOAc, n-BuOH, and water, and 1.39 g of fucosterol was isolated from the dichloromethane fraction [2,43]. Fucosterol was identified using ^1^H and ^13^C-NMR spectroscopy [33,44], and the structure of fucosterol is shown in Table 2.

### 3.2. Eckol

According to the research of Joe et al., ES extract showed an inhibitory effect on the activity of NF-κB, AP-1, and MMP-1, and the compound that exerted the inhibitory effect was identified as eckol and dieckol, which provided the possibility for the development of new anti-photoaging agents [25]. Through HPLC analysis, the content of eckol in the dried ES EtOAc extract was 3.2 ± 0.2 μg/mg [37], while in the ethanol extract of ES, the content of eckol was 12.59 ± 0.52 μg/g [36]. The difference in eckol content may be attributed to the difference in the source of ES, pretreatment method, extraction step, and extraction solvent. Eckol isolated from the ethanolic extract of ES, exhibits a protective effect against doxorubicin-induced hepatotoxicity with an EC_50_ value of 8.3 ± 0.5 μg/mL [34].

### 3.3. Dieckol

Dieckol is an important active ingredient in ES. It has various properties, including tyrosinase inhibition, antioxidant, anti-obesity, anti-diabetic, anti-cancer, anti-photoaging, anti-atherosclerosis, and anti-hepatotoxicity activities [22,24,25,31,34,35,44,45,46]. It has attracted much attention because of its biological activity and has thus been widely used in pharmacology and food. It also protects algae from various stresses in the ecosystem in which the algae live [47]. Preliminary extraction of ES can be performed using acetone, methanol, water, ethanol, EtOAc, butanol, and CH_2_Cl_2_ [16,47]. The dieckol content in the extraction process may vary due to the extraction solvent and ES pretreatment method. Dieckol is an indicator component of ES extract. The content of dieckol in the EtOAc extract of ES was 30.1 ± 0.7 μg/mg [37]. The content of dieckol in 70% ethanol extract of ES was 27.42 ± 0.66 mg/g [38].

### 3.4. Phlorofucofuroeckol A

Phlorofucofuroeckol-A is reported to be present in various brown algae and is a phlorotannin with various biological activities [48], including tyrosinase inhibition and antioxidant, anti-obesity, anti-diabetic, anti-inflammatory, anti-atherosclerotic, anti-hepatotoxic activities [16,17,21,22,24,31,34,35,40,44,49]. Phlorofucofuroeckol-A is a pentacyclic substructure that includes a dibenzo-p-dioxin skeleton (Table 2). It is an edible compound that can be used to treat diseases by injection and oral administration [50,51].

### 3.5. Other Components

Various bioactive components such as phloroglucinol [17,21,24,31], eckstolonol [24,31], 7-phloroeckol [35,49], and dioxinodehydroeckol [22,34] have also been isolated from the ES extract. In the EtOAc grade (4.2 g) methanol extract of ES, 8 mg of 7-phloroeckol was isolated [35]. 7-phloroeckol shows anti-diabetic and antiatherogenic activities [35,49]. Dioxinodehydroeckol, isolated from ES, has been shown to have anti-diabetic and antioxidant activities [22,34]. Fucosterol and 24-hydroperoxy 24-vinylcholesterol were isolated and identified from the n-hexane fraction of ES ethanol extract. Eight phlorotannins, 7-phloroeckol, 2-phloroeckol, triphlorethol-A, dieckol, phlorofucofuroeckol-A, eckol, eckstolonol, and phloroglucinol, were isolated from the EtOAc fraction [18]. Two novel phlorotannins, 974-A [21] and 974-B [16], with a molecular weight of 974, as well as 6,6′-bieckol, 2′-phloroeckol, and phlorofucofuroeckol B, were isolated from the ethanolic extract of ES [16]. A total of 24.5 mg of 974-A was obtained in 63.8 g of the EtOAc fraction of the ethanolic extract [21]. In the LPS-induced inflammation model RAW 264.7 cell experiments, 974-B, 6,6′-bieckol, 2′-phloroeckol, and phlorofucofuroeckol B showed inhibitory effects on NO production, and the EC_50_ of anti-inflammatory activity was 9.72 ± 0.45, 63. 9± 4.51, 85.3 ± 6.1, 12.1 ± 0.79 μmol/L, respectively [16].

## 4. Bioactivities

ES is a representative brown alga of the *Ecklonia* species and is an important food source for some marine animals [1]. In Japan and Korea, it is harvested from the wild and cultivated, and many people eat it as a healthy food [4]. ES has multiple biological activities that are closely related to phlorotannins and marine polyphenols, especially dieckol, phlorofucofuroeckol A, and eckol. It also gives ES multiple health benefits including anti-cancer, anti-obesity, anti-diabetes, antibacterial, and antioxidant effects, among others. (Table 3). The pharmacological activities of ES are summarized as follows. 

### 4.1. Antibacterial Activity

Research and development of natural antibacterial agents is becoming a hot topic in the food industry. Marine algae have attracted much attention because of their richness in bioactive compounds such as phlorotannins and fucoidans [23,59]. Kuda et al. [23] studied the antibacterial properties of two boiled and four dried ES products and found that two of the four dried products showed antibacterial (*Escherichia coli*, *Pseudomonas aeruginosa*, *Corynebacterium glutamicum*, *Staphylococcus aureus*, and *Bacillus cereus*) activity, whereas the two products subjected to boiling treatment had no effect on various bacterial species, and the content and activity of bioactive compounds in ES brown algae were greatly reduced by processing and preservation methods. Boiling resulted in the loss of some phenolic compounds and water-soluble polysaccharides, such as phlorotannins, alginate, and fucoidan. Choi et al. [60] reported that the aqueous extract of ES exerted an inhibitory effect on microbial pathogens. Eom et al. [61] showed that the methanol extract of ES had significant antibacterial activity against methicillin-resistant *Staphylococcus aureus* (MRSA). To conduct a more detailed study of the antibacterial activity of the extract, the methanol extract was further fractionated with organic solvents such as EtOAc, CH_2_Cl_2_, hexane, and n-butanol, and it was determined that the hexane fraction had the strongest antibacterial activity against MRSA strains with minimum inhibitory concentrations ranging from 500 to 600 μg/mL. Lee et al. isolated 87 mg of dieckol in the EtOAc fraction (25.0 g) of ES methanolic extracts and investigated the synergistic effect of dieckol with β-lactams against methicillin-resistant Staphylococcus aureus. The results showed that the minimum inhibitory concentration of dieckol against standard MRSA and methicillin-susceptible *S. aureus* strains was 32–64 μg/mL. Furthermore, when ampicillin and penicillin were used in combination with dieckol (8 and 16 μg/mL), the fractional inhibitory concentration indices for all MRSA strains were 0.066–0.266 μg/mL. It showed that the combination of dieckol and β-lactam has an effective therapeutic effect on MRSA infection [57].

### 4.2. Tyrosinase Inhibition

Tyrosinase is a copper-containing oxidoreductase, also known as polyphenol oxidase, and is widely found in animals, plants, and microorganisms. It directly affects melanin synthesis, acting as the rate-limiting enzyme. Inhibition of tyrosinase activity can prevent skin darkening and melanin production, and tyrosinase inhibitors are widely used in medicine, cosmetics, and food [62,63]. ES is rich in polyphenols and its main component is phlorotannin [64]. Kang et al. [24] isolated five phloroglucinol derivatives from the methanolic extract of ES: dieckol, eckol, eckstolonol, phloroglucinol, and phlorofucofuroeckol A. These five compounds inhibited the oxidation of L-tyrosine catalyzed by mushroom tyrosinase, with IC_50_ values of 2.16, 33.2, 126, 92.8, and 177 μg/mL, respectively. Compared with the tyrosinase inhibitors arbutin and kojic acid, their IC_50_ values were 6.32 and 112 μg/mL, respectively. Using a Lineweaver–Burk plot, it was determined that there were differences in the inhibition modes of the five compounds. Among them, eckstolonol and phloroglucinol were competitive inhibitors, and the inhibition constants (Ki) were 3.1 × 10^−4^ and 2.3 × 10^−4^ M, respectively, while dieckol, eckol, and phlorofucofuroeckol A are noncompetitive inhibitors with Ki of 1.5 × 10^−5^, 1.9 × 10^−5^, and 1.4 × 10^−3^ M, respectively. Manandhar et al. [21] isolated a novel phlorotannin 974-A from ES and simultaneously detected three other known compounds: phlorofucofuroeckol A, eckol, and phloroglucinol. Compound 974-A can inhibit the activities of L-tyrosine and L-DOPA with IC_50_ values of 1.57 ± 0.08 and 3.56 ± 0.22 µM, respectively. Phlorofucofuroeckol-A, eckol, and 974-A downregulated the expression of a tyrosinase-related protein (TRP)-1 and TRP-2, which are involved in melanin production in B16F10 melanoma cells. These results suggest that dieckol, eckol, eckstolonol, phloroglucinol, phlorofucofuroeckol A, and 974-A isolated from ES can be used as potent tyrosinase inhibitors and have potential applications in cosmetics, medicine, and food.

### 4.3. Antioxidant Activity

Antioxidants are substances that effectively inhibit the oxidative reaction of free radicals, thereby of preventing diseases, alleviating human aging, and maintaining the skin [65]. Marine organisms are rich in health value, among which marine algae are the most representative natural antioxidants with high free radical scavenging activity, and are rich in polyphenols, carotenoids, and chlorophyll [66,67]. Kim et al. [22] isolated and identified three phlorotannins (dioxinodehydroeckol, phlorofucofuroeckol A, and dieckol) with high antioxidant activity from the methanol extract of ES using gel, silica gel, and reversed-phase column chromatography, and determined their antioxidant activities by DPPH experiments. The EC_50_ of dioxinodehydroeckol, phlorofucofuroeckol A, and dieckol were 8.8 ± 0.4, 4.7 ± 0.3, and 6.2 ± 0.4 μM, respectively. The antioxidant activities of these three phlorotannins were further determined in vitro, and the results showed that phlorofucofuroeckol A and dieckol inhibited reactive oxygen species (ROS) production in a concentration-dependent manner. Although dioxinodehydroeckol has high DPPH radical scavenging activity, it does not inhibit ROS. This may be because, compared with dioxinodehydroeckol, phlorofucofuroeckol A and dieckol can better penetrate cells and react with ROS, or are more easily involved in the antioxidant system in cells. The difference in antioxidant activity may not only depend on the phenol content of the sample but also on the characteristics of the phenolic compounds contained in the sample, their synergies, and the existence of other chemicals [68,69]. Jun et al. showed that eckol, a phlorotannin isolated from ES ethanolic extract, enhanced heme oxygenase-1 protein and mRNA expression in HepG2 cells by activating c-Jun NH_2_-terminal kinase and PI3K/Akt signaling pathways [55]. Iwai et al. [30] investigated the inhibitory effect of ES on oxidative stress in a diabetic KK-Ay mouse model. In their study, the polyphenol content and free radical scavenging activity of the methanol extract of ES were much higher than those of the water extract, and the active polyphenols in ES extracts were determined to be phlorotannins by LC/MS and HPLC-PDA analysis. Kang et al. [31] determined the inhibitory effect of five compounds (dieckol, phlorofucofuroeckol A, eckol, eckstolonol, and phloroglucinol) in ES extract on total ROS generation, of which phloroglucinol had the highest ability to inhibit total ROS generation with an IC_50_ value of 30.82 ± 2.53 μM. The content and activity of functional compounds in seaweed may vary depending on the preservation conditions and/or processing methods used. Boiling can lead to the loss of the content and activity of the compound because the compound will easily dissolve into the water during the boiling process, or the activity of the compound will be destroyed and reduced at high temperatures [23]. Kuda et al. [23] demonstrated that the total phenolic content and antioxidant activity of ES are affected by a variety of factors, including processing methods and product type. They determined the antioxidant activity of four dried and two boiled ES products, of which both dried and boiled products exhibited lower antioxidant activity. The amount and activity of a compound can be greatly affected by the processing and storage of the product. The active ingredient dieckol contained in the ES extract was nine times more active than the standard butylhydroxytoluene and six times more active than L-ascorbic acid [46].

### 4.4. Anti-Obesity Activity

A clinical trial material made from ES extract significantly inhibited adipogenesis in 3T3-L1 cells and significantly reduced body weight gain, liver weight, white adipose tissue (WAT) weight, and adipocyte size in high-fat diet-induced obese mice. It also improved serum levels of total cholesterol, triglycerides, and LDL cholesterol, and increased HDL cholesterol. Western blot analysis showed that a clinical trial test material made of ES extract reduced the protein expression of C/EBP-α, PPAR-γ, and aP2 to inhibit adipocyte differentiation while promoting the expression of UCP-1 and CPT-1 in adipocytes to increase energy expenditure and play a key role in improving obesity [38]. Jin et al. [70] fed C57BL/6J mice with ES extract at a concentration of 150 mg/kg for six weeks, and the results showed that ES extract significantly reduced body weight gain and WAT weight in mice. In addition, ES extract can promote WAT browning and lipolysis in mice, with excellent anti-obesity effects. Lee et al. [43] isolated the sterol metabolite fucosterol from ES. Fucosterol exhibits an anti-adipogenic ability in vitro, which inhibits adipocyte differentiation and lipid accumulation by downregulating SREBP-1, PPAR-γ, and C/EBP-α, as well as the regulation of multiple signaling pathways (phosphoinositide 3-kinase/Akt and extracellular signal-regulated kinase-dependent forkhead box protein O signaling pathway) that play an anti-obesity role. The anti-adipogenic effect of ES has been demonstrated in a study by Jung et al. [2] in which it was reported that fucosterol, isolated from the methanolic extract of ES, reduced lipid accumulation in a concentration-dependent manner. This study analyzed the active ingredients contained in the ES extract, and the results showed that the ES extract mainly contained five phlorotannins: phlorofucofuroeckol A, dioxinodehydroeckol, eckol, phloroglucinol, and dieckol. Among them, phlorofucofuroeckol A, eckol, and phloroglucinol exhibited concentration-dependent inhibition of lipid accumulation in adipocytes at concentrations ranging from 12.5–100 μM [17]. In a male Sprague–Dawley rat model of alcoholic fatty liver disease, ES extract decreased the expression of the triglyceride synthesis-related gene SREBP-1 and promoted the expression of PPAR-α and CPT-1 fatty acid oxidation-related genes to inhibit fat synthesis. It also reduces the blood levels of alanine aminotransferase, triglycerides, aspartate aminotransferase, and total cholesterol [36].

### 4.5. Anti-Diabetic Activity

Diabetes mellitus is a worldwide epidemic characterized by hyperglycemia, and its prevalence is increasing daily [71]. It is mainly caused by defective insulin secretion, impaired biological action, or both [72]. Long-term hyperglycemia leads to chronic damage and dysfunction of various tissues, especially of the eyes, kidneys, heart, blood vessels, and nerves [73]. Many studies have demonstrated that ES can effectively prevent and treat diabetes and its complications [28,30,44]. Iwai [30] identified ES as an effective diabetes preventive agent based on in vitro and in vivo studies. ES methanol extract containing 303.0 mg/g total polyphenol showed a strong inhibitory effect on α-glucosidase in vitro. In vivo, ES methanol extract was taken orally at 81.20 mg/day and plasma glucose levels were significantly reduced in KK-A Y diabetic mice after 4 weeks. Five phlorotannins, eckol, phlorofucofuroeckol-A, dieckol, phloroglucinol, and dioxinodehydroeckol were isolated from ES ethanolic extract. Among them, phlorofucofuroeckel-A, dieckol, eckol, and dioxinodehydroeckol all showed inhibition of amyloid-β_25–35_ Self-Aggregation at the concentration of 10 μM. In addition, phlorofucofuroeckol-A inhibited D-glucose and D-ribose induced non-enzymatic insulin glycosylation in a dose-dependent manner [58]. Lee and Jeon isolated three phlorotannins 7-Phloroeckol, Dieckol, and phlorofucofuroeckol-A from ES methanolic extracts, which were able to inhibit the activity of the α-glucosidase enzyme [74]. Various phlorotannins such as phlorofucofuroeckol-A, dieckol, and eckol contained in ES had significant inhibitory effects on the activity of angiotensin-converting enzyme with IC_50_ of 12.74 ± 0.15 μm, 34.25 ± 3.56 μm, and 70.82 ± 0.25 μm, respectively [53]. Jung et al. [44] isolated fucosterol from ES and reported that this compound showed moderate inhibitory activity against human recombinant aldose reductase and rat lens aldose reductase and was a mixed-type inhibitor. It was determined by the Autodock program that fucosterol could bind tightly to the enzymatic active site on aldose reductase, thus inhibiting its activity of aldose reductase. Six phlorotannins, 7-phloroeckol, dieckol, phloroglucinol, eckol, phlorofurofucoeckol-A, and dioxinodehydroeckol, were isolated from ES. Among these, 7-phloroeckol, dieckol, and phlorofurofucoeckol-A significantly inhibited α-glucosidase, with IC_50_ values ranging from 1.37 to 6.13 µM. ES and its six phlorotannins also show significant inhibitory activity against protein tyrosine phosphatase 1 B [28].

### 4.6. Anti-Inflammatory Activity

In RAW 264.7, phlorofucofuroeckol A isolated from ES significantly inhibited the inflammatory response induced by lipopolysaccharide (LPS) and reduced the mRNA expression levels of inflammation-related proteins inducible nitric oxide synthase (iNOS) and cyclooxygenase-2 (COX-2), as well as the pro-inflammatory factors interleukin-1β, IL-6, and tumor necrosis factor-α. Phlorofucofuroeckol A also inhibits the activities of nuclear factor-κB (NF-κB) and AP-1 transcription factor promoters [49]. In LPS-stimulated RAW 264.7 cells, the ethanolic extract of ES decreased the expression of NF-κB by inhibiting the degradation of inhibitor κB-α and blocking the phosphorylation of mitogen-activated protein kinases and protein kinase B. Simultaneously, the ethanolic ES extract further downregulated the expression of inflammatory markers (IL-6, iNOS, TNF-α, COX-2, and IL-1β) by inhibiting the NF-κB pathway. The main anti-inflammatory components in the ethanolic extract were identified as phlorofucofuroeckol A and B by HPLC analysis [40]. Kim et al. [22] isolated three phlorotannins, dioxinodehydroeckol, dieckol, and phlorofucofuroeckol A, from ES using a series of column chromatography methods. Among these, only phlorofucofuroeckol A exhibited significant anti-inflammatory activity. Additionally, it further inhibited the production of NO and prostaglandin E2 by inhibiting the protein expression of iNOS and COX-2. In a study by Wei et al. [16], ES was first extracted using ethanol, followed by fractional distillation of the extract. The EtOAc fraction inhibited the production of prostaglandin E2 and nitric oxide in a dose-dependent manner and inhibited the expression of IL-6, TNF-α, and IL-1β pro-inflammatory cytokines. The main anti-inflammatory components in ES were identified by HPLC as 974-B, phlorofucofuroeckol A, 2′-phloroeckol, phlorofucofuroeckol B, and 6,6′-bieckol. Methanol: chloroform (1:2, *v/v*) was used to obtain the ES extract. It inhibits the activities of degranulated enzymes (lipoxygenase and hyaluronidase) in rat basophilic leukemia-2H3 cells in a dose-dependent manner, which may play a role in the treatment of allergic and inflammatory symptoms [56].

### 4.7. Treat Neurological Disorders

Alzheimer’s disease is a degenerative disease of the central nervous system that occurs in old age and pre-senile age and is characterized by progressive cognitive impairment and behavioral impairment and clinically manifested as memory impairment, aphasia, personality, and behavioral changes [54]. Hae et al. isolated eight phlorotannins, 7-phloroeckol, 2-phloroeckol, triphlorethol-A, dieckol, phlorofucofuroeckol-A, eckol, eckstolonol, and phloroglucinol, from the EtOAc fraction of ES ethanolic extract. Two sterols, fucosterol and 24-hydroperoxy 24-vinylcholesterol, were isolated from the n-hexane fraction. The compounds eckstolonol and phlorofucofuroeckol-A both exhibited inhibitory activities against acetylcholinesterase and butyrylcholinesterase with IC_50_ values of 42.66 ± 8.48, 230.27 ± 3.52 μM, and 4.89 ± 2.28, 136.71 ± 3.33 μM, respectively. These results indicated that phlorotannins and sterols contained in ES have potential therapeutic effects in the treatment of Alzheimer’s disease [18]. In a study by Seong et al., four phlorotannins were isolated from ES ethanolic extract: phlorofucofuroeckol-A, dieckol, phloroglucinol, and dioxinodehydroeckol. Among them, phlorofucofuroeckol-A showed the highest inhibitory activity against human monoamine oxidase-A and B, with IC_50_ values of 9.22 ± 0.19 and 4.89 ± 0.32 μm, respectively. In addition, Dieckol and phlorofucofuroeckol-A exhibited a multi-target combination of dopamine D3/D4 receptor agonism and antagonism of dopamine D1/serotonin/neurokinin-1. This suggests that the phlorotannins contained in ES are potential drugs for the treatment of neuronal diseases [15].

### 4.8. Other Functionalities

Dieckol, isolated from ES, reduced the increase in the number of human hepatocellular carcinoma Hep3B cells and induced apoptosis in a dose-dependent manner. Dieckol increases the permeability of the mitochondrial membrane, promotes the release of the apoptotic factor cytochrome c, and induces apoptosis in Hep3B cells [45]. ES extract inhibited the expression of matrix metalloproteinase-1 in human dermal fibroblasts and also had a significant inhibitory effect on NF-κB and AP-1. HPLC analysis identified eckol and dieckol as effective anti-photoaging components in the extract by HPLC analysis [25]. Jun et al. prepared ES gold nanoparticles (GNPs) by mixing ES extract (2 mg/mL) with 1 mM gold (III) chloride solution with vigorous stirring and then incubating at 25 °C for 15 min. The antisenescence effect of ES-GNPs in a human dermal fibroblasts (HDF) senescence model induced by ultraviolet A (UVA) irradiation was investigated. The results showed that ES-GNPs exerted antisenescence effects on UVA-irradiated HDF by inhibiting MMP-1/-3 expression and SA-β-galactosidase activity [75]. In addition, fucosterol extracted from ES down-regulated the expression of glucose-regulated protein 78 to alleviate soluble amyloid beta peptide_1–42_-induced cognitive impairment in aging rats [33]. In vitro, ES extract (1000 µg/mL) significantly reduced phenylephrine-induced cardiomyocyte hypertrophy and directly inhibited the activity of p300-HAT as well as the acetylation of histone H3K9. In in vivo experiments, treatment of mice with myocardial infarction with ES extract showed that ES significantly inhibited the increase in myocardial cell diameter, vascular fibrosis, and histone H3K9 acetylation [9]. The ES formulation can significantly inhibit liver injury induced by carbon tetrachloride in rats, promote an increase in antioxidant enzymes, and inhibit the activity of CYP2E1 in a dose-dependent manner [76]. The ethanolic extract of ES (163.93 μg/mL) showed inhibitory properties on angiotensin-converting enzyme with an inhibition rate of more than 50% [77]. The methanol extract of ES and four phlorotannins (7-phloroeckol, dieckol, phlorofucofuroeckol A, and eckol) isolated from ES effectively inhibited the Cu2+-induced oxidation of low-density lipoprotein [35]. Fucosterol, isolated from the leafy thalli of ES, significantly ameliorated the increase in ROS levels and decrease in glutathione levels induced by tert-butyl hydroperoxide and tacrine [52]. Vo et al. isolated a novel phlorotannin fucofuroeckol-A (F-A) from ES extracts—the chemical structure is shown in Table 1—and investigated the protective effect of FA on ultraviolet B (UVB)-induced allergic reactions in RBL-2H3 mast cells. They found that F-A (50 μM) could prevent mast cell degranulation by promoting the elevation of intracellular Ca^2+^ and inhibiting the release of histamine and showed inhibitory effects on the production of TNF-α and IL-1β, which could effectively prevent the occurrence of allergic reactions caused by UVB [19].

## 5. Literature Search

Literature searches were performed using multiple databases, including Google Scholar, PubMed, MDPI, Elsevier, Hindawi, and the National Center for Biotechnology Information. The relevant literature on ES published from 1990 to 2022 was searched for a review of the bioactive components and bioactivities of ES. The keywords used in the literature search were “Ecklonia stolonifera” or “Bioactive” or “Compound” or “bioactivity,” “Antibacterial” or “Tyrosinase inhibition” or “Antioxidant” or “Anti-obesity” or “Anti-diabetic” or “Anti-inflammatory” or “Treat neurological disorders” or “anti-photoaging” or “antisenescence” or “phlorotannin”. The articles related to the keywords were selected from the retrieved articles to summarize the ES. Through the review results of ES, the limitation that most studies have focused on the crude extract of ES or one or several components in the extract was found, and related research on the content changes of ES contained in the whole growth stage has not been well mentioned.

## 6. Conclusions

ES is rich in bioactive components, has various health benefits, and can be used as a potential functional food or drug to prevent and treat various diseases. The purpose of this review is to summarize the bioactivities of ES and the various bioactive compounds it contains to facilitate the development and application of ES in food, medicine, and clinical fields. The bioactive compounds involved were fucosterol, 24-hydroperoxy 24-vinylcholesterol, phloroglucinol, eckol, eckstolonol, triphlorethol-A, dieckol, dioxinodehydroeckol, phlorofucofuroeckol A, phlorofucofuroeckol B, fucofuroeckol-A, 7-phloroeckol, 2′-phloroeckol, 6, 6′-bieckol, 974-A, and 974-B. These bioactive compounds endow ES with various bioactive potentials, such as anti-obesity, anti-diabetic, anti-Alzheimer, tyrosinase inhibition, antioxidant, anti-photoaging, anti-atherosclerotic, anti-hepatotoxic, treat neurological disorders, anti-inflammatory, etc. Although ES is being studied more widely, there are still many shortcomings: the bioactive components contained in ES may vary with cultivars, growing environments, and stages of growth and maturity. In addition, the known biologically active compounds contained in ES may have additional health benefits, and there may still be some unknown compounds with various biological activities. There are also many deficiencies in the related studies of ES in clinical trials. Therefore, the unidentified bioactive compounds in ES and the biological effects should be investigated in depth in future studies, and consistent phytochemical profiles as well as suitable drug delivery systems should also be identified to facilitate future clinical trials and development and research in pharmacological applications.

## Figures and Tables

**Table 1 marinedrugs-20-00607-t001:** The main extraction methods of ES and their bioactive compounds.

Active Compound	Extract Type	Source	Content (μg/g)	References
Fucosterol	Methanol extract	Lyophilized powder	11,920.0	[2,33]
	Ethanol extract	Lyophilized powder	10,752.7	[18]
24-hydroperoxy 24-vinylcholesterol	Ethanol extract	Lyophilized powder	1792.1	[18]
Phloroglucinol	Methanol extract	Lyophilized powder	240.0	[34]
			-	[24]
	Ethanol extract	Air-dried powder	-	[21]
		Lyophilized powder	3920.0	[18]
Dioxinodehydroeckol	Methanol extract	Air-dried powder	-	[22]
		Lyophilized powder	60.0	[35]
	Ethanol extract	Dry powder in direct sunlight	-	[16]
Eckol	Methanol extract	Lyophilized powder	280.0	[35]
			-	[24]
	Ethanol extract	Air-dried powder	-	[21]
			12.0–13.0	[36]
		Lyophilized powder	5400.0	[18]
		Dry powder in direct sunlight	-	[16]
	Hexane Extract	Lipid-removed dried powder	3000.0–3400.0	[37]
Phlorofucofuroeckol A	Methanol extract	Air-dried powder	-	[22]
		Lyophilized powder	150.0	[35]
			-	[24]
	Ethanol extract	Air-dried powder	-	[21]
		Lyophilized powder	2280.0	[18]
	Hexane Extract	Lipid-removed dried powder	11.3–13.5	[36]
			7200.0–8200.0	[37]
Dieckol	Methanol extract	Air-dried powder	-	[22]
		Lyophilized powder	1260.0	[35]
			-	[24]
	Ethanol extract	Air-dried powder	145.0–149.0	[36]
		Fresh ES	26,760.0–28,080.0	[38]
		Dry powder in direct sunlight	-	[16]
		Lyophilized powder	3480.0	[18]
	Hexane Extract	Lipid-removed dried powder	29,400–30,800.0	[37]
7-phloroeckol	Methanol extract	Lyophilized powder	70.0	[35]
	Ethanol extract	Lyophilized powder	800.0	[18]
2-phloroeckol	Ethanol extract	Dry powder in direct sunlight	-	[16]
		Lyophilized powder	360.0	[18]
Phlorofucofuroeckol B	Ethanol extract	Dry powder in direct sunlight	-	[16]
6,6′-bieckol	Ethanol extract	Dry powder in direct sunlight	-	[16]
974-A	Ethanol extract	Air-dried powder	122.5	[21]
974-B	Ethanol extract	Dry powder in direct sunlight	-	[16]
Eckstolonol	Methanol extract	Lyophilized powder	-	[24]
Triphlorethol-A	Ethanol extract	Lyophilized powder	2400.0	[18]

**Table 2 marinedrugs-20-00607-t002:** Bioactive compounds, structures, and molecular formulas contained in ES.

Bioactive Compounds	Molecular Formula	Structure	References
Fucosterol	C_29_H_48_O	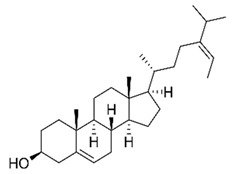	[34,39]
Phloroglucinol	C_6_H_6_O_3_	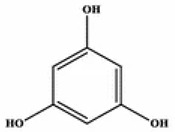	[17,18,21,24,34]
Eckol	C_18_H_12_O_9_	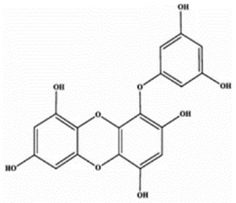	[16,18,21,24,35,36,37]
Eckstolonol	C_18_H_10_O_9_	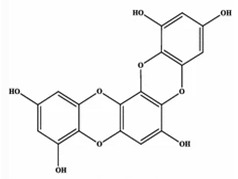	[24,31]
Dieckol	C_36_H_22_O_18_	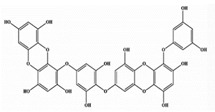	[16,18,22,24,35,36,37,38]
Dioxinodehydroeckol	C_18_H_10_O_9_	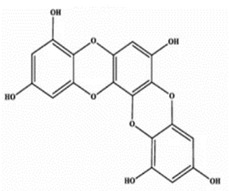	[16,22,34,35]
Phlorofucofuroeckol A	C_30_H_18_O_14_	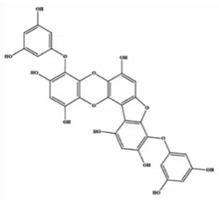	[18,21,24,36,37]
Phlorofucofuroeckol B	C_30_H_18_O_14_	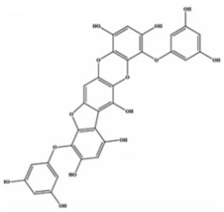	[16,40]
7-phloroeckol	C_24_H_16_O_12_	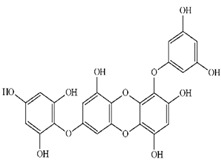	[18,35]
2′-phloroeckol	C_24_H_16_O_12_	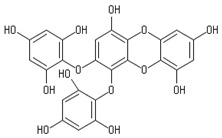	[16,18]
6,6′-bieckol	C_36_H_22_O_18_	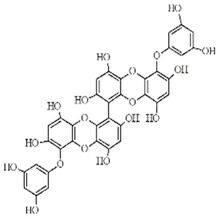	[16]
974-A	C_48_H_30_O_23_	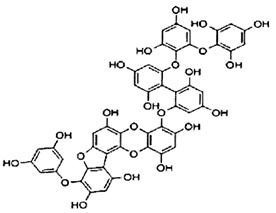	[21]
974-B	C_48_H_30_O_23_	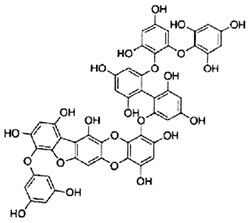	[16]

**Table 3 marinedrugs-20-00607-t003:** Important bioactive compounds from ES and their bioactivities.

Bioactive Compounds	Biological Activity	Major Findings	References
Fucosterol	Anti-diabetic	Inhibit the activity of aldose reductase.	[44]
	Anti-hepatotoxic	Ameliorate the increase in ROS levels and decrease in glutathione levels	[52]
	Cognitive impairment improvement	Down-regulates the expression of glucose-regulated protein 78	[33]
	Anti-obesity	Inhibits adipocyte differentiation and lipid accumulation	[2,43]
Phloroglucinol	Anti-diabetic	Inhibit the activity of protein tyrosine phosphatase 1B	[28]
	Anti-obesity	Inhibit lipid accumulation	[17]
	Antioxidant	Inhibit ROS generation	[31]
	Tyrosinase inhibition	Inhibition of L-tyrosine activity	[24]
		Inhibit the activities of L-tyrosine and L-DOPA	[21]
Eckol	Anti-atherosclerotic	Inhibit low-density lipoprotein oxidation	[35]
	Anti-photoaging	Inhibit the expression of matrix metalloproteinase 1	[25]
	Anti-diabetic	Inhibits amyloid-β25-35 self-aggregation	[53]
		Inhibit the activity of protein tyrosine phosphatase 1B	[28]
		Inhibit angiotensin converting enzyme activity	[54]
	Anti-obesity	Inhibit lipid accumulation	[17]
	Antioxidant	Enhanced heme oxygenase-1 protein and mRNA expression	[55]
		Inhibit ROS generation	[31]
	Tyrosinase inhibition	Inhibition of L-tyrosine activity	[24]
		Inhibit the activities of L-tyrosine and L-DOPA	[21]
Eckstolonol	Treat neurological disorders	Inhibit acetylcholinesterase and butyrylcholinesterase activity	[18]
	Antioxidant	Inhibit ROS generation	[31]
	Tyrosinase inhibition	Inhibition of L-tyrosine activity	[24]
Dieckol	Anti-atherosclerotic	Inhibit low-density lipoprotein oxidation	[35]
	Anti-photoaging	Inhibit the expression of matrix metalloproteinase 1	[25]
	Anti-cancer	Promotes the release of the apoptotic factor cytochrome c	[45]
	Anti-obesity	Inhibits adipocyte differentiation and lipid accumulation	[38]
	Treat neurological disorders	Inhibit the activity of human monoamine oxidase-A and B	[15]
	Anti-diabetic	Inhibits amyloid-β25-35 self-aggregation	[53]
		Inhibit the activity of the α-glucosidase enzyme	[28,56]
		Inhibit the activity of protein tyrosine phosphatase 1B	[28]
		Inhibit angiotensin-converting enzyme activity	[54]
	Antibacterial	Anti-MRSA	[57]
	Antioxidant	DPPH free radical scavenging activity	[22]
		Inhibit ROS generation	[31]
	Tyrosinase inhibition	Inhibition of L-tyrosine activity	[24]
Dioxinodehydroeckol	Antioxidant	DPPH free radical scavenging activity	[22]
	Anti-diabetic	Inhibits amyloid-β25-35 self-aggregation	[53]
		Inhibit the activity of protein tyrosine phosphatase 1B	[28]
Phlorofucofuroeckol A	Anti-atherosclerotic	Inhibit low-density lipoprotein oxidation	[35]
	Treat neurological disorders	Inhibit acetylcholinesterase and butyrylcholinesterase activity	[18]
		Inhibit the activity of human monoamine oxidase-A and B	[15]
	Anti-inflammatory	Inhibit the activity of inflammation-related proteins (iNOS, TNF-α, COX-2, IL-6, IL-1β, and NF-κB, AP-1)	[22,40,49]
		Inhibit the activity of degranulated enzymes (lipoxygenase and hyaluronidase)	[58]
	Anti-diabetic	Inhibits amyloid-β25-35 self-aggregation	[53]
		Inhibit the activity of the α-glucosidase enzyme	[28,56]
		Inhibit the activity of protein tyrosine phosphatase 1B	[28]
		Inhibit angiotensin-converting enzyme activity	[54]
	Anti-obesity	Inhibit lipid accumulation	[17]
	Antioxidant	DPPH free radical scavenging activity	[22]
		Inhibit ROS generation	[31]
	Tyrosinase inhibition	Inhibition of L-tyrosine activity	[24]
		Inhibit the activities of L-tyrosine and L-DOPA	[21]
Phlorofucofuroeckol B	Anti-inflammatory	Inhibit the activity of inflammation-related proteins (iNOS, TNF-α, COX-2, IL-6, IL-1β, and NF-κB, AP-1)	[40]
		Inhibit the activity of degranulated enzymes (lipoxygenase and hyaluronidase)	[58]
Fucofuroeckol-A	Anti-allergy	Inhibit the production of TNF-α and IL-1β	[19]
7-phloroeckol	Anti-atherosclerotic	Inhibit low-density lipoprotein oxidation	[35]
	Anti-diabetic	Inhibit the activity of the α-glucosidase enzyme	[28,56]
		Inhibit the activity of protein tyrosine phosphatase 1B	[28]
2′-phloroeckol	Anti-inflammatory	Inhibit the activity of degranulated enzymes (lipoxygenase and hyaluronidase)	[58]
6,6′-bieckol	Anti-inflammatory	Inhibit the activity of degranulated enzymes (lipoxygenase and hyaluronidase)	[58]
974-A	Tyrosinase inhibition	Inhibit the activities of L-tyrosine and L-DOPA	[21]
974-B	Anti-inflammatory	Inhibit the activity of degranulated enzymes (lipoxygenase and hyaluronidase)	[58]

## Data Availability

The data presented in this study are available within this article.

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
