# Peer review of "In-Depth Understanding of Ecklonia stolonifera Okamura: A Review of Its Bioactivities and Bioactive Compounds"

_marinedrugs, 2022, doi:10.3390/md20100607_

Round 1

Reviewer 1 Report

It was my pleasure to read the narrative review “In-Depth Understanding of Ecklonia stolonifera: A Review of its Bioactive Compounds and Bioactivities”. The manuscript focuses on the potential health benefits of bioactive constituents of the brown seaweed Ecklonia stolonifera (ES).  Algae represent an important new source of different molecules capable of exerting a wide range of biological effects that could be exploited in medicine, nutrition, and many other fields, so I believe it is useful to review the bioactivities of a given species of algae.  However, I suggest a major revision of this work prior to acceptance.

Main comments to the authors:

My main concern is some confusion in the organization of the paper. I suggest the following:

1) In the introduction section you should merge all the information on ES that are now scattered in the text (environmental distribution, physical characteristics, chemical composition, use as food or drug, and so on…).  

2) In several paragraphs yields of different compounds are reported according to the extraction method. I think this is an important point that deserves more attention. I suggest reviewing the extraction methods used to obtain ES  crude extracts and isolated compounds, and discuss them along with the importance of different environments, seasonality, reproductive status, etc. to determine the final content of different bioactive molecules. You may decide to organize this issue in a dedicated paragraph.

 3) Lines 69-70 “Further fractionation and separation of the methanolic ES resulted in 1.39 g of fucosterol in previous studies”: this sentence makes no sense here. I think you are referring to the yield of fucosterol from ES, but you need to explain better and rephrase (this is related to my previous comment).

 4) Then I would go through the bioactivities (anti-microbial, anti-thyrosinase, and so on…) of both crude extracts and isolated compounds (namely different phlorotannins).

 5) You can proceed with a Discussion section, in which you propose a critical review of the results, and Table 1 as a summary of results. You might consider adding to the Table the results obtained from several ES crude extracts, or you could separate tables. For a critical review consider, for example: what was the contribution of in vivo studies with respect to in vitro models? What are the future specific research needs? What about dietary intake, absorption and bioavailability of bioactive compounds (especially phlorotannins)? What about the use of dietary supplements? Is there a preventive potential besides a therapeutic one? And so on….

 6) A short Conclusion section should put an end to your work.

 Minor comments

 1) The paper is generally well written and clear enough, but there are some grammar errors and typos, so authors are advised for a language revision.

2) The first time you mention it, please indicate the full name of the alga according to the base https://www.algaebase.org.

3) I would change the title:  In-Depth Understanding of Ecklonia stolonifera: A Review of its Bioactivities and Bioactive Compounds.

 4) After the full revision of the manuscript, check the Abstract and rewrite if necessary.

 Author Response

Reviewer 2 Report

Line 52, 83 and Table 1-I suggest to replace 'anti-ageing' by 'anti-photoageing' since the cited reference No. 17 is about anti-photoageing.

Table 1-Molecular structures drawing sholud be more uniform and clearer; dieckol crushed structure is the worse example.

Line 137-'dimethyl chloride' is possibly 'dichloromethane' ?

2.2. Eckol-it should be added its anti-photoageing activity as in the Table 1.

In Chapeter 3, Bioactivities, there is missing information about anti-photoageing activity.

Reviewer 4 Report

The authors reviewed bioactive compounds and their biological activity from seaweed Ecklonia stolonifera. In my opinion, it’s an interesting review. The subject is interesting and in line with the current direction of marine research, especially algal research. The paper is well written and surprisingly easy to read.  I would suggest considering this manuscript for publication after major revision.

As a general comment, the authors should give some methodological information regarding the literature search. How did they proceed with the literature search? Did they do systematic research? If yes, which databases did the authors use? Pubmed, Scopus, WOS, others? And which keywords? What is the range of the years or other criteria used for the review? Only 9 out of 63 references are recent, why? If there is no research on the topic this needs to be discussed.

In my opinion, all parts need to be revised by adding a critical scientific opinion and discussion. In the current form, only fact listing is presented.

The conclusion needs to be rewritten in the way that major findings and future perspectives are described, not general comments.

Some specific comments:

Table 1- please fit structure photos in lines so that they don't interfere

Line 115 – name of order should be in italics

Line 175 – 50 in EC50 shouldn’t be superscript

Round 2

Reviewer 1 Report

The authors performed a major revision of their work, which is now well organized and more complete. In my opinion the paper is now acceptable for publication on Marine Drugs.

Author Response

We appreciate you for the time and effort on our paper. 

Reviewer 3 Report

The MS was revised, and the authors ansered all the questions, generally the MS was improved greatly.

Still, there are some blunts as followed;

1. The author mentioned in the abstract on that ES is rich in polysaccharides, but not given in the MS. 

2. Ref 15. reported the early acetate fraction without ethanol extact, please check the Table 1 and all the references.

Author Response

Dear reviewer,

We appreciate you for the time and effort on our paper. The responses to reviewer comments have been marked in blue throughout the manuscript. We hope the changes have addressed all the shortcomings outlined. Below you could find our point-by-point response to the reviewer comments.

Point 1: The author mentioned in the abstract on that ES is rich in polysaccharides, but not given in the MS.

Response 1: Thanks for your suggestions very much. We have added a supplementary note regarding polysaccharides contained in ES: “Through gel filtration column and high performance liquid chromatography (HPLC) analysis, Kuda et al. determined that the water extract of ES contained some low molecular weight polysaccharides, and determined that these polysaccharides were laminarin [7]. The results of sugar composition analysis indicated that the types of polysaccharides contained in ES cell walls were cellulose, fucoidan, and laminaran [10]. “ Please see lines 51-56.

Point 2: Ref 15. reported the early acetate fraction without ethanol extact, please check the Table 1 and all the references.

Response 2: Thanks for your suggestions very much. We have changed the sentence of “Wei et al. dried the collected ES under direct sunlight for 2 days, extracted the powder with 95% ethanol, and then fractionated with n-hexane, EtOAc, and n-BuOH solution. Eight phlorotannins, phlorofucofuroeckol A, dieckol, 974-B, 6,6′-bieckol, phlorofuco-furoeckol B, eckol, dioxinodehydroeckol, and 2-phloroeckol were isolated from the ethyl acetate fraction [15].” to “Wei et al. dried the collected ES under direct sunlight for 2 days, extracted the powder with 95% ethanol, and then dissolved the obtained powder extract with methanol. Eight kinds of phlorotannins were isolated from it, which were phlorofucofuroeckol A, dieckol, 974-B, 6,6′-bieckol, phlorofucofuroeckol B, eckol, dioxinodehydroeckol, and 2-phloroeckol [16].”. Please see lines 109-113.

Reviewer 4 Report

I thank the authors for considering all the comments and making the necessary changes to the paper. I find it significantly improved and acceptable for publication.

Author Response

(The authors gave the same response as above.)
